# Greedy Equivalence Search in the Presence of Latent Confounders

**Tom Claassen**[1]                    **Ioan Gabriel Bucur**[1]

[1]Institute for Computing and Information Sciences, Radboud University, Nijmegen, (The) Netherlands

## Abstract

We investigate Greedy PAG Search (GPS) for score-based causal discovery over equivalence classes, similar to the famous Greedy Equivalence Search algorithm, except now in the presence of latent confounders. It is based on a novel characterization of Markov equivalence classes for MAGs, that not only improves state-of-the-art identification of Markov equivalence between MAGs to linear time complexity for sparse graphs, but also allows for efficient traversal over equivalence classes in the space of all MAGs. The resulting GPS algorithm is evaluated against several existing alternatives and found to show promising performance, both in terms of speed and accuracy.

## 1 INTRODUCTION

Ever since the advent in the early 90's of modern, principled methods for causal discovery from observational data, there have been two main paradigms that have been widely employed: constraint-based and score-based methodologies. Both start from the assumption that there is some underlying causal structure, typically in the form of a directed acyclic graph (DAG), that is responsible for the observed data distribution. The first class of methods then search for (conditional) in/dependence constraints between variables in the data, and use this information in combination with certain orientation rules to reconstruct the output causal model. Key assumptions include the *causal Markov assumption*, essentially stating that the structure of the underlying graph induces independence constraints in the observed data according to the *d*-separation criterion (see below), as well as the *causal faithfulness assumption*, stating that these are also the only observable independencies in the data. Other simplifying model assumptions like acyclicity and causal sufficiency (no latent confounders) can also be employed.

When causal sufficiency does not apply the target causal model can be represented as a (maximal) ancestral graph (MAG, see below). The output then represents the so-called Markov equivalence class (MEC) of the underlying causal model, in the form of a partial ancestral graph (PAG) representing all causal graphs that satisfy the same independence model. Benchmark examples of algorithms in this tradition include PC and FCI (Spirtes et al., 2000), where the latter is sound and complete even in the presence of latent confounders and selection bias.

In contrast, score-based approaches define a metric that quantifies how well a certain graph structure captures the observed data, and then iteratively try to search for a graph that maximizes this score. The score is typically based on a (Bayesian) likelihood in combination with a penalty on model complexity, and usually assumes an underlying DAG structure with no unobserved confounders. A classic example is the K2 algorithm by Cooper and Herskovits (1992),

In many cases, it is possible to choose a score in such a way that all graphs in the same equivalence class obtain the same score (Heckerman et al., 1995). As there can be a huge number of graph instances in the same equivalence class, this opens up the possibility of significantly speeding up the search by moving between equivalence classes rather than between individual graphs. This was the motivation behind algorithms like GBPS (Spirtes and Meek, 1995), and its famous successor GES (Greedy Equivalence Search) (Chickering, 2002b), as well as recent versions improving scaling behaviour and statistical efficiency (Ramsey et al., 2017; Chickering, 2020). In practice, equivalence search significantly outperforms traditional graph based search methods, both in speed and accuracy. Due to the global nature of the score, their output also tends to be more robust than that of their constraint-based counterparts. Unfortunately, like PC, they also assume causal sufficiency, meaning that there is currently no available method that can employ the full potential of score-based equivalence search in the presence of latent confounders. Addressing this gap is the focus of this article.

*Accepted for the 38th Conference on Uncertainty in Artificial Intelligence* (UAI 2022).

**Towards equivalence search for MAGs**

There have been several related score-based methods in recent years that try to go beyond the standard DAG search. For example Triantafillou and Tsamardinos (2016) consider the relative performance of constraint-based methods vs. MAG search using the BIC score for multivariate Gaussian distributions from (Richardson and Spirtes, 2002). Their GS-MAG algorithm employed a greedy search over the space of MAGs, where at each step all possible single edge modifications were evaluated. Later results showed this could be improved by starting from the MMPC skeleton Tsirlis et al. (2018). GSMAG was found to have promising performance, albeit at much greater running times.

A different approach was taken by Ogarrio et al. (2016). They managed to circumvent the MAG equivalence search by exploiting the original GES to first do equivalence search in the space of DAGs, and then to add a post-processing step using a modification of FCI that started from the GES output in order to obtain the final PAG. The result was a hybrid method (GFCI, short for Greedy FCI)) that showed promising performance over either method separately, but did not exploit the potential of full PAG search.

Bhattacharya et al. (2021) presented a radically different alternative that tackles the even wider class of ancestral ADMGs by exploiting differentiable algebraic constraints to turn causal discovery in a continuous optimization problem.

In the meantime many transformational characterizations of MAGs have been developed, see e.g. (Tian, 2012; Zhang and Spirtes, 2012), showing that we can reach all MAGs within the same equivalence class by a series of (covered) edge reversals to go from one MAG to the next where all are part of the same MEC. But as these characterizations are primarily concerned with transformations *within* the same equivalence class, they are not easy to generalize into an orthogonal search strategy *between* equivalence classes.

Our solution to this problem is based on a novel MEC characterization for MAGs that does not rely on complicated paths but on straightforward collider/noncollider triples. Any change to these triples implies a new MEC, which makes it easy to generate a collection of neighbouring MECs. In combination with an appropriate score this then forms the main engine in our Greedy PAG Search (GPS) algorithm for score-based equivalence search in the presence of latent confounders.

The rest of the article is organised as follows: section 2 introduces some basic concepts and terminology, section 3 describes the new characterization for Markov equivalence between MAGs, section 4 discusses how to use this for traversal between equivalence classes in the MAG space, ultimately leading to the GPS algorithm in section 5. Section 6 then shows the performance of GPS in practice compared to some state-of-the-art alternatives.

## 2 NOTATION AND TERMINOLOGY

A *mixed graph* $\mathcal{G}$ is a graphical model that can contain three types of edges between pairs of nodes: directed ($\rightarrow$), bidirected ($\leftrightarrow$), and undirected ($-$). In a mixed graph, standard graph-theoretical notions, e.g. *child/parent*, *ancestor/descendant*, *directed path, cycle*, still apply, with natural extension to sets. A vertex $z$ is a *collider* on a path $\pi = \langle \ldots, x, z, y, \ldots \rangle$ if there are arrowheads at $z$ on both edges from $x$ and $y$, otherwise it is a *noncollider*. A triple $x - z - y$ on a path is *unshielded* if $x$ and $y$ are not adjacent in $\mathcal{G}$. An unshielded collider is known as *v-structure*.

A mixed graph $\mathcal{G}$ is *ancestral* iff an arrowhead at $x$ on an edge to $y$ implies there is no directed path from $x$ to $y$ in $\mathcal{G}$, and there are no arrowheads at nodes with undirected edges. As a result, arrowhead marks can be read as 'is not an ancestor of'. In a mixed graph $\mathcal{G}$, a vertex $x$ is *m-connected* to $y$ by a path $\pi$, relative to a set of vertices $Z$, iff every noncollider on $\pi$ is not in $Z$, and every collider on $\pi$ is an ancestor of $Z$. If there is no such path, then $x$ and $y$ are *m-separated* by $Z$. An ancestral graph is *maximal* (MAG) if for any two nonadjacent vertices there is a set that m-separates them. A *directed acyclic graph* (DAG) is a special kind of MAG, containing only $\rightarrow$ edges, for which *m*-separation reduces to the standard *d*-separation criterion. For more details, see (Koller and Friedman, 2009; Spirtes et al., 2000).

A *causal DAG* $\mathcal{G}_C$ is a directed acyclic graph where the arcs represent direct causal interactions (Pearl, 2009). In general, the independence relations between observed variables in a causal DAG can be represented in the form of a MAG (Richardson and Spirtes, 2002). The (complete) partial ancestral graph (PAG) represents all invariant features that characterize the equivalence class $[\mathcal{G}]$ of such a MAG, with a tail '$-$' or arrowhead '$>$' end mark on an edge, iff it is invariant in all $[\mathcal{G}]$, otherwise a circle mark '$\circ$'.

## 3 CHARACTERIZING MARKOV EQUIVALENCE CLASSES

In this section we introduce a modified characterization for the Markov equivalence class (MEC) of MAGs, that will form the basis for the equivalence search in the next section. It also leads to a simple method to establish Markov equivalence between MAGs.

### 3.1 MECS OF MAGS

For Markov equivalence between MAGs we start from the following characterization from Ali et al. (2009):

**Lemma 1** *Two MAGs $\mathcal{G}_1$ and $\mathcal{G}_2$ belong to the same Markov equivalence class if and only if they have the same skeleton and the same colliders with order.*

This reflects the well known characterization for DAGs where two members are in the same equivalence class iff they have the same skeleton and *v*-structures, with the latter now generalized to 'collider triples with order':

**Definition 1** *Let $\mathfrak{T}_i(i \geq 0)$ be the set of triples of order $i$ in a MAG $\mathcal{G}$, defined recursively as:*

- *A triple $\langle a, b, c \rangle \in \mathfrak{T}_0$ if $a \ast{-}\ast b \ast{-}\ast c$ is in $\mathcal{G}$, with $a$ and $c$ not adjacent.*
- *A triple $\langle a, b, c \rangle \in \mathfrak{T}_{i \geq 1}$ if $\langle a, b, c \rangle \notin \mathfrak{T}_{j<i}$, and there is a discriminating path $\langle x, q_1, .., q_p, a, b, c \rangle$ for $b$ in $\mathcal{G}$ (possibly $q_1 = a$), where the $p + 1$ colliders $\langle x, q_1, q_2 \rangle, ..., \langle q_{p-1}, q_p, a \rangle, \langle q_p, a, b \rangle \in \bigcup_{j<i} \mathfrak{T}_j$.*

Here a path $\pi = \langle x, q_1, .., q_p, a, b, c \rangle$ in $\mathcal{G}$ is a *discriminating path* for $b$ iff $x$ is not adjacent to $c$, and every vertex between $x$ and $b$ is a collider along $\pi$ and is parent of $y$ in $\mathcal{G}$. For example in Figure 1, $\langle A, B, C, E \rangle$ would be a discriminating path for $C$, and $\langle A, B, C, D, E \rangle$ for $D$.

Note that triples $\langle a, b, c \rangle$ and $\langle c, b, a \rangle$ are equivalent, and that triples with order $i \geq 1$ are triangles in $\mathcal{G}$. Also note that the final condition is only needed to uniquely determine the order $i$, but that the characterization itself does not depend on the actual value. This characterization leads to an algorithm for testing Markov equivalence between two MAGs with polynomial complexity $O(ne^4)$, with $n$ the number of vertices and $e$ the number of edges in the graph.

More recently, Hu and Evans (2020) came up with a characterization in terms of a parameterizing set $\mathcal{S}_3(\mathcal{G})$ based on so-called *heads* and *tails* of the districts (connected bidirected components) in $\mathcal{G}$, and the '3' indicates only sets of up to 3 nodes are required. In contrast with (Ali et al., 2009) it does *not* rely on the discriminating path, and leads to an even more efficient algorithm for checking equivalence that runs in $O(ne^2)$ for sparse graphs (when $n = O(e)$). Unfortunately, this characterization is difficult to translate into a comprehensive search strategy between equivalence classes.

However, it turns out that we can also circumvent the discriminating path in Definition 1 in another way.

## 3.2 A NEW 'TRIPLES WITH ORDER' CHARACTERIZATION

On closer inspection of the second part of Definition 1 we see that every discriminating path (see Figure 1) can be viewed as a collection of collider and noncollider triples with order. More importantly, to know that a path $\langle x, q_1, .., q_p, z, y \rangle$ is a valid discriminating path for $z$ in $\mathcal{G}$ it suffices to know that $\langle x, q_1, .., q_{p-1}, q_p, y \rangle$ is a valid discriminating path for noncollider $q_p$ along the path, and that $\langle q_{p-1}, q_p, z \rangle$ is a collider, and that $z$ and $y$ are adjacent in

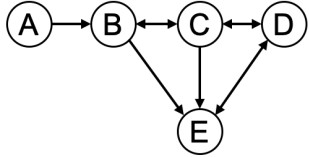

Figure 1: MAG with discriminating paths A-B-C-(D)-E.

| $k$ | $\mathfrak{C}$ | | |
|---|---|---|---|
| 0 | A | B | C |
| 0 | B | C | D |
| 0 | B | E | D |
| 2 | C | D | E |

| $k$ | $\mathfrak{D}$ | | |
|---|---|---|---|
| 0 | A | B | E |
| 1 | B | C | E |

Table 1: Corresponding 'triples with order' lists.

$\mathcal{G}$. But that also means we do not actually need the full discriminating path, but we just need to know that $\langle q_{p-1}, q_p, y \rangle$ is a noncollider with order, and that $\langle q_{p-1}, q_p, z \rangle$ is a collider with order. This results in the following alternative characterization:

**Definition 2** *Let $\mathfrak{C}_i$ resp. $\mathfrak{D}_i$ $(i \geq 0)$ be the set of collider- resp. noncollider triples with order $i$ in a MAG $\mathcal{G}$, defined recursively as:*

- *A triple $\langle a, b, c \rangle \in \mathfrak{C}_0$ (resp. $\mathfrak{D}_0$), if $a \ast{-}\ast b \ast{-}\ast c$ is an unshielded collider (resp. noncollider) in $\mathcal{G}$.*
- *A triple $\langle a, b, c \rangle \in \mathfrak{C}_i$ (resp. $\mathfrak{D}_i$), with $i \geq 1$, if $\langle a, b, c \rangle \notin \mathfrak{C}_{j<i}$ (resp. $\mathfrak{D}_{j<i}$), and*
    1. *$a \ast{-}\ast b \ast{-}\ast c$ is a collider (noncollider) in $\mathcal{G}$,*
    2. *$\exists q : \langle q, a, b \rangle \in \mathfrak{C}_{j<i}$, and $\langle q, a, c \rangle \in \mathfrak{D}_{k<i}$.*

The connection to the original 'triple with order' definition follows from the next lemma (proof in the supplement):

**Lemma 2** *In a MAG $\mathcal{G}$, a triple $\langle a, b, c \rangle$ is in $\mathfrak{C}_i$ (resp. $\mathfrak{D}_i$), if and only if $\langle a, b, c \rangle \in \mathfrak{T}_i$ and $\langle a, b, c \rangle$ is a collider (resp. noncollider) in $\mathcal{G}$.*

This motivates the following definition:

**Definition 3** *The MEC $\mathcal{M}$ of a MAG $\mathcal{G}$, denoted $\mathcal{M}(\mathcal{G})$, is defined as the triplet $\langle \mathcal{S}, \mathfrak{C}, \mathfrak{D} \rangle$, with $\mathcal{S}$ the (undirected) skeleton of $\mathcal{G}$, and $\mathfrak{C}$ and $\mathfrak{D}$ the corresponding lists of collider resp. noncollider triples with order from Definition 2.*

Which leads to the straightforward implication:

**Corollary 3** *Two MAGs $\mathcal{G}_1$ and $\mathcal{G}_2$ are Markov equivalent if and only if $\mathcal{M}(\mathcal{G}_1) = \mathcal{M}(\mathcal{G}_2)$.*

From here on we will use the term MEC to denote this particular representation of the Markov equivalence class of a MAG $\mathcal{G}$.

## 3.3 FROM MAG TO MEC

Definition 2 implies that after we established the unshielded (non)collider triples with order 0, we only need to check the already constructed lists and a specific (non)collider triple in the graph $\mathcal{G}$ in order to identify each higher order triple. This leads to the following **MAG-to-MEC** procedure:

---

**Algorithm 1** MAG-to-MEC

---
**Input:** MAG $\mathcal{G}$
**Output:** MEC $\{\mathcal{S}, \mathfrak{C}, \mathfrak{D}\}$
*phase 1: initialise, process unshielded triples*
$\mathcal{S} \leftarrow Skeleton(\mathcal{G})$
$\mathfrak{C}_0/\mathfrak{D}_0 \leftarrow$ unshielded (non)colliders $\langle x, z, y \rangle \in \mathcal{G}$
**for all** $\langle x, z, y \rangle \in \mathfrak{D}_0$ **do**
   **if** $\exists q : \langle x, z, q \rangle \in \mathfrak{C}_0$ and $\mathcal{G}(q, y) > 0$ **then**
      $\mathfrak{L} \leftarrow \langle z, q, y \rangle$ {*initialise process list $\mathfrak{L}$*}
   **end if**
**end for**
*phase 2: process candidate triples until no more left*
**repeat**
   $\langle x, z, y \rangle \leftarrow Pop(\mathfrak{L})$
   **if** $x \ast\rightarrow z \leftarrow\ast y$ in $\mathcal{G}$ **then**
      add $\langle x, z, y \rangle$ to $\mathfrak{C}$
      $\forall q : \langle x, z, q \rangle \in \mathfrak{D}, \mathcal{G}(q, y) > 0$: add $\langle z, y, q \rangle$ to $\mathfrak{L}$
   **else**
      add $\langle x, z, y \rangle$ to $\mathfrak{D}$
      $\forall q : \langle x, z, q \rangle \in \mathfrak{C}, \mathcal{G}(q, y) > 0$: add $\langle z, q, y \rangle$ to $\mathfrak{L}$
   **end if**
**until** $\mathfrak{L}$ is empty
**return** $\mathcal{S}, \mathfrak{C}, \mathfrak{D}$

---

Algorithm 1 gives a high-level overview of the corresponding steps. (implementation details available at `https://github.com/tomc-ghub/gps_uai2022`) It starts by identifying all unshielded triples (order 0) and allocating them to the appropriate collider or noncollider lists. After that, all triples with order 1 are collected in list $\mathfrak{L}$, and processed one by one depending on whether they correspond to a collider or noncollider in the graph. Each allocated triple may give rise to new triples with order that are added to the end of the list $\mathfrak{L}$, until we have found them all. For each processed triple (allocated to $\mathfrak{C}$ or $\mathfrak{D}$) we only need to consider the existence of matching triples in the complementary list together with the presence of a specific edge in the MAG to find the new implied (higher order) triples. Table 1 shows the output $\mathfrak{C}$ and $\mathfrak{D}$ lists given the MAG in Figure 1.

## 3.4 FROM MEC BACK TO MAG

For the reverse **MEC-to-MAG** direction we can directly map all triples with order into specific (minimal) edge mark orientations to obtain the so-called **core PAG** (Definition 4), and then propagate the remaining implied orientations

using a subset of the standard FCI orientation rules from Zhang (2008) to obtain the completed PAG. From there we can obtain a matching MAG instance by following, e.g. the arc-augmentation procedure in Theorem 2 of (Zhang, 2008) which will result in a fully oriented MAG in the same MEC with a minimum number of (invariant) bidirected and undirected edges.

**Definition 4** *(core PAG)* *For a MEC $\mathcal{M} = \langle \mathcal{S}, \mathfrak{C}, \mathfrak{D} \rangle$, the core PAG $\mathcal{P}^*$ is defined as the graph obtained from the skeleton $\mathcal{S}$ with all $\circ\!-\!\circ$ edges, in combination with*

- $\forall \langle x, z, y \rangle \in \mathfrak{C}_0$   : *orient* $x \ast\rightarrow z \leftarrow\ast y$ *in* $\mathcal{P}^*$
- $\forall \langle x, z, y \rangle \in \mathfrak{C}_{k \geq 1}$ : *orient* $z \leftarrow\ast y$ *in* $\mathcal{P}^*$
- $\forall \langle x, z, y \rangle \in \mathfrak{D}_{k \geq 1}$ : *orient* $z -\!\ast y$ *in* $\mathcal{P}^*$

Each collider with order 0 becomes a *v*-structure, and each triple with order $k \geq 1$ corresponds to exactly one invariant edge mark (arrowhead or tail) in the graph. Note that in processing triples $\langle x, z, y \rangle$ with order $k \geq 1$, we rely on the fact that they are stored in the lists such that the $y$ entry corresponds to the final node in a discriminating path, which is easily done when constructing the MEC.

The justification for the notion of a'core PAG' is that the resulting graph contains all invariant information needed to uniquely establish the full, completed PAG, by only propagating the graphical FCI orientation rules, i.e. *without* the need for specific independence test results as required by *v*-structure rule $\mathcal{R}0$ and the discriminating path rule $\mathcal{R}4$ in (Zhang, 2008).

However, it is possible that an invariant tail is hiding in a higher order noncollider triple that is not a 'triple with order', hence we need one more rule to ensure completeness. As it fulfils much the same role as the original $\mathcal{R}4$, we will refer to it as rule $\mathcal{R}4'$:

$\mathcal{R}4'$: Let $Z$ be a district among the parents of a node $y$. If $x \ast\rightarrow z \longrightarrow y$, with $z \in Z$ and $x$ and $y$ not adjacent, then orient all $u \ast\rightarrow y$ with $u \ast\rightarrow z'$ for some $z' \in Z$ (possibly $z' = z$) as $u \longrightarrow y$.

---

**Algorithm 2** MEC-to-CPAG

---
**Input:** MEC $\{\mathcal{S}, \mathfrak{C}, \mathfrak{D}\}$
**Output:** completed PAG $\mathcal{P}$
$\mathcal{P} \leftarrow \mathcal{P}^*(\mathcal{S}, \mathfrak{C}, \mathfrak{D})$    (*the core PAG from Definition 4*)
run orientation rules $\mathcal{R}1 - \mathcal{R}4'$ on $\mathcal{P}$  (*all arrowheads*)
run orientation rules $\mathcal{R}5 - \mathcal{R}10$ on $\mathcal{P}$ (*remaining tails*)
**return** $\mathcal{P}$

---

The following lemma ensures the output is indeed sound and complete:

**Lemma 4** *For a valid MEC $\mathcal{M}$, algorithm 2 will output the corresponding completed PAG $\mathcal{P}$.*

## 3.5 ALGORITHMIC COMPLEXITY

Checking for Markov equivalence between MAGs simply corresponds to building the MEC for one, and verifying that the same steps apply to the other. This will induce a constant cost for each entry in the MEC, and so the algorithmic complexity for increasing graph sizes is determined by the complexity of building the MEC from a given MAG.

To estimate the worst-case time complexity of algorithm 1 consider graphs over $n$ nodes with $e$ edges and max. node degree $d$. For sparse graphs with $d \leq k$ we have $e = O(n)$, whereas in general we can have $e = O(n^2)$.

The first phase of the algorithm requires finding all unshielded triples, which means selecting all pairs of nodes from the neighbours of every node in the graph, leading to $n \cdot d \cdot (d-1) = O(nd^2)$ triples. For the initialization of the temporary triple list $\mathfrak{L}$ we need to check all triples $\langle x, y, z \rangle$ in $\mathfrak{C}_0$, and compare with specific entries in the complementary list $\mathfrak{D}_0$ (or vice versa) for nodes adjacent to $z$ in $\mathcal{G}$. With appropriate indexing that implies an additional $d$ candidates to check for each entry in the smaller of the two lists, bringing the total for phase 1 to $O(nd^3)$.

Each entry in the temporary list is then processed and compared against $d$ other candidates, each of which can be handled in constant time as it involves only verifying presence in one of the (non)collider triple lists, which can again be done in constant time using appropriate indexing, and the presence of a specific edge in $\mathcal{G}$, also in constant time. Each combination added corresponds to a triangle in the graph, meaning there are at most $O(nd^2)$ triples to process, where each requires checking $d$ entries, again leading to a combined total of $O(nd^3)$ steps for phase 2.

Together that means for sparse graphs we have worst case linear complexity of $O(n)$ (!), whereas in general this leads to $O(n^4)$. This is actually a significant improvement over the $O(ne^2)$ complexity reported by Hu and Evans (2020), corresponding to $O(n^3)$ for sparse graphs and $O(n^5)$ for arbitrary density (when $e = O(n^2)$).

These complexity results relate to the worst-case scaling behaviour, and in practice the typical performance may scale much better. For example the empirical complexity for sparse graphs in Hu and Evans (2020) seemed much closer to our linear result, meaning that in practice the two characterizations may be expected to perform similarly (see section 6.1). The main contribution of our new representation therefore lies in the way it enables us to traverse the MEC/PAG space in the next section.

## 4 MOVING BETWEEN PAGS

The main goal in this article is to find a search strategy that allow us to move directly from one equivalence class to

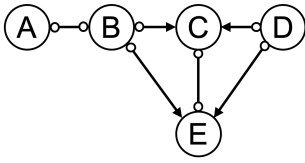

Figure 2: PAG after $MakeNoncollider(A, B, C)$ on Figure 1.

another, as the basis for an iterative (greedy) score-based causal discovery algorithm, similar in spirit to GES for DAGs (Chickering, 2002a), but now in the presence of latent confounders. For that we need a principled way to generate a set of new candidate neighbouring equivalence classes from a given starting equivalence class. Key aspects here are deciding *what* to change, and then *how* to change it, in order to ensure the resulting target corresponds to a different but valid equivalence class.

For the 'what', the new characterization in terms of the MEC $\mathcal{M}$ provides a natural starting point, as any change by definition leads to a new equivalence class. This suggests the following basic operators:

- **AddEdge** - insert an edge between two nodes in $\mathcal{S}$,
- **DeleteEdge** - remove a single edge from $\mathcal{S}$,
- **MakeNoncollider** - move a triple $\langle x, z, y \rangle$ in $\mathfrak{C}$ to $\mathfrak{D}$,
- **MakeCollider** - move a triple $\langle x, z, y \rangle$ in $\mathfrak{D}$ to $\mathfrak{C}$.

However, this does not fully answer the 'how' yet. A single application on a MEC of one of the operators above can lead to many implied changes, creating as well as destroying other 'triples with order'. For example, in Figure 1, turning collider triple $\langle A, B, C \rangle$ into a noncollider would imply the destruction of both higher order triples in Table 1, leading to the PAG in Figure 2. At the same time, not all operators can act in isolation, e.g. if two or more triples share an edge in the PAG, and some changes may be invalid, e.g. if it would introduce an invariant arrowhead at a node on an undirected edge in the PAG. To avoid such inconsistencies and recognise which triples should be modified in conjunction, we implement the operators to act directly on the PAG $\mathcal{P}$.

That leaves the problem of how to convert the resulting modified graph into a valid equivalence class, as the standard FCI orientation rules do not suffice given that certain invariant edge marks in the starting PAG may no longer be invariant in the target PAG. Fortunately, here too the new MEC characterization comes to the rescue. On closer inspection we see that we could equally well use a PAG as input for the MAG-to-MEC procedure in Algorithm 1, or indeed the modified graph $\mathcal{G}$ resulting from applying an operator on the PAG $\mathcal{P}$.

The main challenge that remains is that an operator may introduce a level of ambiguity through newly created triples with order that are *not* fully determined by the modified

graph $\mathcal{G}$. For example, given the PAG in Figure 2, executing the reverse operator $MakeCollider(A, B, C)$ implies that $\langle B, C, E \rangle$ is a new triple with order, but we have no information on whether it should be a collider or a noncollider, and indeed both options would lead to a valid PAG.

In the baseline implementation of our operators, below, we resolve this ambiguity by constructing a specific instance for the $Add/DeleteEdge$ operators that is guaranteed to be valid, and choosing a default 'noncollider' option for all remaining undetermined higher order triples.

Having reconstructed the modified MEC $\mathcal{M}'$ we can use Algorithm 2 to obtain the corresponding PAG $\mathcal{P}'$, and expand to an (arc-augmented) MAG instance $\mathcal{G}'$, This MAG can subsequently be used to validate the output. The resulting procedure is depicted in Algorithm 3.

---

**Algorithm 3** PAG Candidate Neighbours

   **Input:** MEC $\mathcal{M}$, PAG $\mathcal{P}$, active Operators
   **Output:** collection of PAG $\{Neighbours\}$
   **for all** active Operators, target edges/triples in $\mathcal{M}$ **do**
      $\mathcal{G} \leftarrow Operator(\mathcal{P}, target)$    *(modified graph)*
      $\mathcal{M}' \leftarrow MAG\_to\_MEC(\mathcal{G})$   *(rebuild MEC, Alg.1)*
      $\mathcal{P}' \leftarrow MEC\_to\_PAG(\mathcal{M}')$  *(expand)*
      $\mathcal{G}' \leftarrow PAG\_to\_MAG(\mathcal{P}')$   *(arc-augmentation)*
      **if** $IsValidMAG(\mathcal{G}')$ **then**
         $Neighbours\{end + 1\} \leftarrow \{\mathcal{M}', \mathcal{P}', \mathcal{G}'\}$  *(add)*
      **end if**
   **end for**
   **return** $\{Neighbours\}$

---

### 4.1 BASELINE OPERATOR IMPLEMENTATION

The potential for inconsistencies and ambiguity from applying operators arbitrarily to a PAG means that a form of validation is necessary to ensure we obtain meaningful candidate neighbour PAGs at each step of the algorithm.

Unfortunately, determining a sound and complete expansion of a PAG with arbitrary background information (the modified graph $\mathcal{G}$ in algorithm 3) is still an open problem. In combination with the potential ambiguity from undetermined higher order triples that means we cannot (yet) provide a full a priori 'if and only if' validity check for all of the operators. However, we can incorporate some basic checks to ensure we do not try candidates that will lead to obvious inconsistencies. In the experimental results in section 6 we will find that this already filters out the vast majority of invalid candidate PAGs

The validity checks for the the *MakeCollider* and *MakeNoncollider* operators simply verify the edge marks in the modified graph $\mathcal{G}$ would not violate the definition of a valid MAG (no arrowheads at nodes on undirected edges, and no (almost) directed cycles). The *MakeNoncollider* operator is

the most involved, as there we need to consider multiple versions to create a noncollider triple, possibly leading to multiple, different output PAGs: three versions for order 0 colliders: $x \ast\!\!-\ z$, $z -\!\!\ast y$, and $x \ast\!\!-\ z -\!\!\ast y$, and one for higher order: $z \longrightarrow y$ The *AddEdge* and *DeleteEdge* operators are constructed such that they are always valid, and so do not require an explicit validity check prior to execution. Both are based on a tail-augmented MAG instance (Zhang, 2006) of the source PAG $\mathcal{P}$, which needs to be derived only once per iteration.

$AddEdge(\mathcal{P}, x, y)$: Let $\mathcal{G}$ be a tail-augmented MAG instance of $\mathcal{P}$. If both $x$ and $y$ have no arrowheads in $\mathcal{G}$, then add $x -\!\!- y$ to $\mathcal{G}$. Otherwise, if $x$ does not have an arrowhead in $\mathcal{G}$, but $y$ does, then add $x \longrightarrow y$ (or v.v.). Otherwise, if $x \in An(y)$ in $\mathcal{G}$ then add $x \longrightarrow y$, if $y \in An(x)$ then add $x \longleftarrow y$, if neither then add $x \longleftrightarrow y$ to $\mathcal{G}$.

$DeleteEdge(\mathcal{P}, x, y)$: Let $G$ be a tail-augmented MAG instance of $\mathcal{P}$. Remove edge $x \ast\!\!-\!\!\ast y$ from $\mathcal{G}$.

$MakeCollider(\mathcal{P}, x, z, y)$: Check for no other $u -\!\!- z$ in $\mathcal{P}$. Let $\mathcal{G}$ be the graph from setting $x \ast\!\!\rightarrow z \leftarrow\!\!\ast y$ in $\mathcal{P}$. Check there is no (almost) directed cycle in $\mathcal{G}$ involving $x$, $y$ and $z$.

$MakeNoncollider(\mathcal{P}, x, z, y)$: (Order 0, version 1): If not exists collider triple $u \ast\!\!\rightarrow z \leftarrow\!\!\ast y$ then skip (=equivalent to version 3). Check not $x \longrightarrow z$ in $\mathcal{P}$ (arrowhead at undirected edge), check $x \longleftarrow z$ would not be part of an (almost) directed cycle. Create $\mathcal{G}$ by setting $x \ast\!\!-\ z$ in $\mathcal{P}$. (Version 2): Idem for $z -\!\!\ast y$. (Order 0, version 3): Check if $x \longrightarrow z$, then no $u \ast\!\!\rightarrow x$ in $\mathcal{P}$; idem if $y \longrightarrow z$ then no $u \ast\!\!\rightarrow y$; if either then also check no $u \ast\!\!\rightarrow z$. Create $\mathcal{G}$ by setting $x \ast\!\!-\ z -\!\!\ast y$ in $\mathcal{P}$. If $x \longleftarrow z$ in $\mathcal{P}$ then check it is not part of an (almost) directed path in $\mathcal{G}$. Idem for $z \longrightarrow y$. (Higher order): Check $z \longrightarrow y$ would not be part of an almost directed cycle in $\mathcal{P}$. Create $\mathcal{G}$ by setting $z \longrightarrow y$ in $\mathcal{P}$.

In principle the four operators suffice to traverse the entire MEC/PAG space, although that is naturally no guarantee the optimal model will be found in a greedy search strategy.

## 5 GREEDY PAG SEARCH

Given the procedure to obtain different neighbouring PAGs/MECs, all that remains to turn this into an effective search algorithm is a means to score individual PAGs. For simplicity, we will assume a multivariate Gaussian model.

### 5.1 SCORING PAGS

When moving between equivalence classes, algorithm 3 expands each PAG to an arc-augmented MAG instance to verify validity. Given that for multivariate Gaussian models Richardson and Spirtes (2002) already introduced a well-established MAG score, we will rely on that as an associated score for the corresponding equivalence class. Because it

is already part of the literature, we will relegate the description of the Gaussian MAG score to Appendix C in the supplement. For details see also (Nowzohour et al., 2017; Triantafillou and Tsamardinos, 2016).

Note that the GPS algorithm itself is in no way restricted to multivariate Gaussian distributions. For example, we could equally well have chosen the score for binary/discrete data developed in (Drton and Richardson, 2008), or alternatively the ADMG score for nested Markov models in (Shpitser et al., 2013) as a MAG corresponds to an ADMG for a nested Markov model without implied Verma constraints.

However, as it is also known that the Gaussian MAG score can be notoriously unstable for graphs with larger districts, we will also include an evaluation based on the so-called structural Hamming distance (SHD) relative to the true PAG, to illustrate the potential of the GPS search itself, separate from any potential scoring issues.

## 5.2   THE BASELINE GPS ALGORITHM

Having developed all the necessary tools we can now put them together into the (baseline) Greedy PAG Search (GPS) algorithm below. It starts from an empty model and, using the operators from section 4.1, each time greedily tries to find a different, neighbouring PAG that will improve the score the most, until no more improvements can be found.

---

**Algorithm 4** Greedy PAG Search

**Input:** Gaussian covariance $\Sigma$ over $N$ variables
**Output:** optimal matching PAG $\mathcal{P}$, top score $s$
Initialise: $\mathcal{M} \leftarrow$ empty MEC over $N$ variables, $s \leftarrow 0$
**repeat**
  $\{\mathbf{M}\} \leftarrow Candidate\_Neighbours(\mathcal{M})$
  **for all** $\mathcal{M}_i \in \mathbf{M}$ **do**
    $s_i \leftarrow Score(\mathcal{M}_i)$
    **if** $s_i > s$ **then** $(\mathcal{M}, s) \leftarrow (\mathcal{M}_i, s_i)$
  **end for**
**until** no more improvement
**return** $\mathcal{P} \leftarrow MEC\_to\_PAG(\mathcal{M}), s$

---

The baseline PAG search aims to find a single, unambiguous target for each version of the operators. This limits the number of candidates to consider at each iteration in the search, which helps to speed up the overall process. Downside is that it becomes easier to get stuck in local optima, leading to suboptimal final solutions. Therefore we will also consider an alternative GPS version.

## 5.3   EXTENDED GPS SEARCH

Effectively, the baseline operators avoid ambiguity by treating remaining circle marks in the modifed graph as signifying 'noncollider'. But, by definition, for any circle mark

in a PAG there is at least one MAG instance that contains an arrowhead, and so for a newly created unshielded triple involving circles it is possible that the same operator applied to a MAG in the starting $\mathcal{P}$ would have produced an unshielded collider triple instead. And for multiple such instances, any different combination of collider and noncollider triples with order corresponds to a different PAG. For example in Figure 2, removing edge $C \circ\!\!\rightarrow E$ would create two new triples with order 0: $\langle C, B, E \rangle$ and $\langle C, D, E \rangle$, where both could become either collider or noncollider in one of four different valid PAGs.

That means that the current baseline search effectively only considers a small proportion of the possible set of neighbouring PAGs at each step. Therefore we also introduce a version of the search that generates an extended *collection* of neighbours for each operator, one for each possible (non)collider combination of newly introduced unshielded triples. In this version, both *AddEdge* and *DeleteEdge* now start from the PAG (rather than a specific tail-augmented MAG instance), where *AddEdge* also considers all possible edge types to add at each application.

This extended approach is similar in spirit to GES (Chickering, 2002b), that at each step also considers a (potentially large) collection of neighbouring equivalence classes per operator, whereas the baseline search is more in line with (Chickering, 2002a). To avoid the risk of having to consider too many candidates in cases where we encounter dense graphs we simply put a reasonable limit (in our case: 64) on the maximum number of local candidates per operator to consider, again similar to GES. No additional validity checks were implemented per operator, so we may expect the rejection rate to rise compared to the baseline version.

The added rigour of the extended search comes at a noticeable penalty cost in terms of time per iteration. Therefore, as a way of illustrating the flexibility of the approach, we will also consider a *hybrid* version that uses the baseline search as standard, and only switches to the extended version once it gets stuck.

One could envisage similar adaptations that restrict what operators can be used in different search stages, e.g. first only allowing 'AddEdge', and then a second stage that only uses 'DeleteEdge', to mimic the GES strategy. Alternatively, we could start from the output graph found by another method like FCI, and then try to tweak this for further improvements, or restrict the search to stay within a likely skeleton, etc.

Finally, for this article we will only consider single runs for each GPS instance, but other familiar strategies to improve the final output, like tabu-search, multiple restarts, simulated annealing etc. could also be employed. Establishing what ultimately works best in what circumstances will be left as future work.

# 6 EXPERIMENTAL EVALUATION

## 6.1 MAG-TO-MEC COMPLEXITY

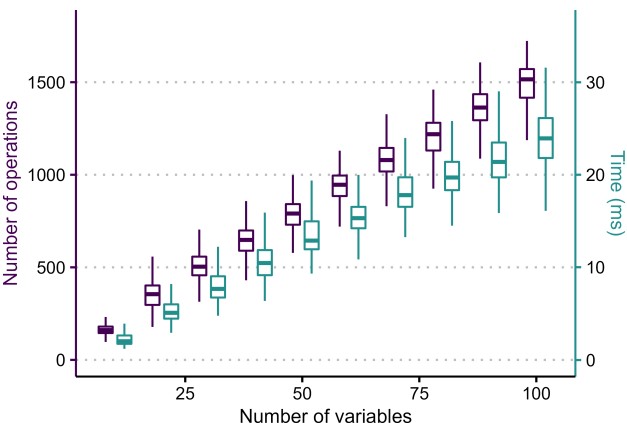

Figure 3: Empirical complexity **MAG-to-MEC**.

A crucial part of the proposed methodology is the new MEC characterization in terms of 'triples with order'. In Section 3.2, we derived that for sparse graphs the theoretical complexity of the **MAG-to-MEC** algorithm is $O(n)$. Figure 3 confirms this via the empirical complexity on random MAGs of size $n = \{10, 20, ..., 100\}$, each averaged over 250 graphs. Similar to the simulation in Hu and Evans (2020) the MAGs are generated to have approximately $e = 3n$ edges, (corresponding to $d = 6$), while each edge is (independently) either directed or bidirected with probability $p = 0.5$.[1] The results demonstrate a strong linear trend (even slightly better), both in terms of 'elementary operations' (purple) and raw computational time (cyan).

## 6.2 GPS SIMULATION EXPERIMENTS

We evaluate the speed and accuracy of the three versions of the **GPS** algorithm. We compare our method against the GS-MAG algorithm proposed by Triantafillou and Tsamardinos (2016) and the GFCI algorithm proposed by Ogarrio et al. (2016), while also showing the results obtained with FCI as a baseline. We also compared against DCD (Bhattacharya et al., 2021), which we found to perform slightly worse than GFCI at significantly longer running times, and so is left out of the final comparison. We generated 100 MAGs for each graph size $n \in \{5, 10, 15, 20\}$, such that the average node degree was $d = 3$, the maximum node degree was $d_{\max} = 10$, and the probability of an edge being bidirected (as opposed to directed) was $p = 0.2$.

We used the following metrics to evaluate the algorithm performance: 1. the Structural Hamming Distance (*SHD*),

---

[1]Simulation details are available with the software at `https://github.com/tomc-ghub/gps_uai2022`.

counting the number of different edges and/or edge marks between the output PAG and the ground truth PAG; 2. the Bayesian information criterion (*BIC*) score for MAGs as proposed by Triantafillou and Tsamardinos (2016); and 3. the *accuracy* of edge marks, obtained as a Jaccard similarity coefficient, by dividing the number of correct edge marks in the output PAG by the total number of edge marks in the (skeleton) union of output and ground truth PAG.

The accuracy results are summarized in Table 2. For all GPS versions and GSMAG, we considered two different starting points for the greedy search, namely the empty graph and the PAG obtained by running the FCI algorithm. We used the BIC score for MAGs (Triantafillou and Tsamardinos, 2016) as the objective function in the greedy optimization. We ran FCI and GFCI using the Tetrad library (Glymour et al., 2014) with default parameters, where Fisher's $z$-test was used for finding conditional independences, and the BIC score was used for the score-based component of GFCI.

In Table 2 we first note that in terms of accuracy when using the BIC score, baseline GPS and GFCI are the clear winners. Extended/hybrid GPS and GSMAG all manage to obtain better (lower) BIC scores, however at abysmal accuracy ratings, indicating a fundamental issue with the Gaussian MAG score. On closer inspection this turns out to result from unstable BIC scores, primarily related to larger districts, where the RICF fitting step fails to converge properly. Baseline GPS tends to favour graphs with fewer/smaller districts (due to the 'default noncollider' option) for which this issue is much less pronounced. However, using the SHD scores shows a dramatically different result: here GSMAG clearly outcompetes the baseline GPS search with accuracy % in the low 90s rather than mid 80s for the latter. However this also demonstrates the potential and effectiveness of the extended search obtaining accuracies of 97-98%. A second interesting observation here is that starting from the FCI PAG actually hinders the extended GPS search from achieving its optimal score by a significant margin (around 10% worse), suggesting that FCI tends to favour a local optimum from which it can be difficult to escape to the optimal graph. This is also reflected in the hybrid version, that runs extended on top of the baseline output, but also is pushed in a certain region of PAG space from which it is harder to escape in single run greedy search.

When it comes to speed, shown in Table 3, baseline GPS arrives at results much faster than GSMAG in all cases, as it needs to consider far fewer candidates per step. Starting from the FCI PAG cuts the number of iterations roughly in half, although time required using the BIC score can actually increase, again signalling the convergence issue. The extended GPS-SHD version shows that the number of iterations required to obtain the optimal model is about a quarter lower than for the baseline version, indicative of the added flexibility the extended neighbour collection can bring. The unsurprising drawback of this larger collection

Table 2: Algorithm accuracy comparison

| Algorithm | GPS baseline | | | | GPS extended | | | | GPS hybrid | | | | GSMAG | | | | GFCI | FCI |
| --- | --- | --- | --- | --- | --- | --- | --- | --- | --- | --- | --- | --- | --- | --- | --- | --- | --- | --- |
| Criterion | BIC | | SHD | | BIC | | SHD | | BIC | | SHD | | BIC | | SHD | | N/A | N/A |
| n metric | empty | FCI | empty | FCI | empty | FCI | empty | FCI | empty | FCI | empty | FCI | empty | FCI | empty | FCI | N/A | N/A |
| SHD (5) | 9.73 | 9.56 | 1.76 | 1.11 | 12.53 | 11.14 | 0.31 | 0.47 | 10.49 | 10.56 | 0.92 | 0.80 | 9.73 | 8.60 | 1.34 | 1.61 | 10.36 | 10.64 |
| BIC (5) | 12.88 | 12.81 | 13.09 | 12.89 | 12.46 | 12.53 | 12.89 | 12.89 | 12.63 | 12.61 | 12.94 | 12.90 | 12.41 | 12.55 | 12.95 | 12.90 | 12.99 | 13.05 |
| accuracy (5) | 0.50 | 0.50 | 0.88 | 0.93 | 0.36 | 0.41 | 0.98 | 0.97 | 0.45 | 0.44 | 0.93 | 0.94 | 0.52 | 0.55 | 0.92 | 0.90 | 0.45 | 0.42 |
| SHD (10) | 24.48 | 23.36 | 5.82 | 5.15 | 35.80 | 31.13 | 0.54 | 3.75 | 31.26 | 28.99 | 2.94 | 4.51 | 38.45 | 31.02 | 3.47 | 2.63 | 21.51 | 22.77 |
| BIC (10) | 30.33 | 30.57 | 32.35 | 31.75 | 28.93 | 29.17 | 31.29 | 31.53 | 28.82 | 29.06 | 31.98 | 31.74 | 28.92 | 28.75 | 31.44 | 31.37 | 31.72 | 31.73 |
| accuracy (10) | 0.49 | 0.50 | 0.83 | 0.84 | 0.33 | 0.38 | 0.98 | 0.88 | 0.40 | 0.41 | 0.91 | 0.86 | 0.32 | 0.42 | 0.90 | 0.92 | 0.48 | 0.45 |
| SHD (15) | 34.15 | 38.47 | 7.26 | 8.42 | 54.90 | 53.32 | 1.53 | 6.75 | 50.19 | 50.21 | 4.59 | 7.67 | 62.90 | 53.64 | 3.60 | 3.58 | 29.99 | 34.10 |
| BIC (15) | 36.54 | 36.48 | 40.29 | 39.63 | 33.24 | 33.51 | 38.59 | 39.25 | 32.58 | 33.42 | 39.91 | 39.61 | 32.83 | 32.31 | 38.19 | 38.18 | 38.74 | 39.18 |
| accuracy (15) | 0.52 | 0.47 | 0.85 | 0.83 | 0.32 | 0.34 | 0.97 | 0.86 | 0.37 | 0.38 | 0.90 | 0.84 | 0.31 | 0.38 | 0.92 | 0.92 | 0.50 | 0.43 |
| SHD (20) | 44.35 | 49.69 | 9.96 | 11.20 | 82.98 | 74.49 | 1.69 | 8.67 | 69.11 | 70.45 | 6.63 | 10.73 | 94.06 | 74.46 | 4.47 | 3.34 | 36.82 | 42.72 |
| BIC (20) | 59.91 | 60.14 | 64.77 | 63.94 | 55.44 | 55.08 | 62.87 | 63.42 | 54.77 | 55.96 | 64.30 | 63.94 | 54.52 | 54.23 | 62.61 | 62.55 | 63.53 | 63.90 |
| accuracy (20) | 0.55 | 0.50 | 0.84 | 0.83 | 0.30 | 0.36 | 0.97 | 0.87 | 0.38 | 0.38 | 0.89 | 0.83 | 0.29 | 0.39 | 0.93 | 0.95 | 0.55 | 0.46 |

Table 3: Algorithm speed comparison

| Algorithm | GPS baseline | | | | GPS extended | | | | GPS hybrid | | | | GSMAG | | | |
| --- | --- | --- | --- | --- | --- | --- | --- | --- | --- | --- | --- | --- | --- | --- | --- | --- |
| Criterion | BIC | | SHD | | BIC | | SHD | | BIC | | SHD | | BIC | | SHD | |
| n metric | empty | FCI | empty | FCI | empty | FCI | empty | FCI | empty | FCI | empty | FCI | empty | FCI | empty | FCI |
| iterations (5) | 7.80 | 2.08 | 9.00 | 3.81 | 6.77 | 2.20 | 8.05 | 3.47 | 9.33 | 3.49 | 10.18 | 4.88 | 8.28 | 3.00 | 8.09 | 3.53 |
| time (s) (5) | 0.33 | 0.18 | 0.23 | 0.15 | 0.97 | 0.49 | 1.12 | 0.67 | 0.63 | 0.45 | 0.52 | 0.38 | 1.60 | 1.24 | 0.53 | 0.48 |
| iterations (10) | 19.31 | 7.34 | 20.41 | 7.56 | 17.73 | 7.45 | 16.29 | 6.80 | 23.37 | 10.99 | 22.20 | 8.76 | 21.14 | 10.72 | 16.94 | 8.34 |
| time (s) (10) | 8.25 | 9.23 | 4.13 | 2.36 | 29.53 | 24.75 | 21.20 | 12.22 | 17.86 | 22.09 | 9.95 | 5.62 | 50.66 | 55.06 | 10.04 | 14.30 |
| iterations (15) | 27.53 | 13.18 | 29.68 | 11.06 | 26.86 | 13.57 | 23.12 | 9.20 | 34.69 | 19.74 | 31.49 | 12.23 | 33.04 | 17.98 | 25.07 | 12.69 |
| time (s) (15) | 33.62 | 55.08 | 18.25 | 9.88 | 156.62 | 187.08 | 102.49 | 51.07 | 98.08 | 149.59 | 41.05 | 20.24 | 321.62 | 360.39 | 70.83 | 59.20 |
| iterations (20) | 38.65 | 18.47 | 39.79 | 13.40 | 40.10 | 21.36 | 30.82 | 11.27 | 48.85 | 28.49 | 41.74 | 14.53 | 48.92 | 26.60 | 33.14 | 15.96 |
| time (s) (20) | 107.39 | 167.10 | 56.80 | 28.00 | 564.51 | 729.35 | 365.16 | 173.52 | 342.30 | 489.25 | 113.12 | 51.15 | 926.26 | 863.41 | 240.71 | 148.44 |

is that the actual running time can be 5-6x greater. Hybrid GPS performance is somewhere halfway between the two.

To give an indication of the effectiveness of the validity checks: on a typical batch of 16 graphs over various sizes and densities we found that baseline GPS rejected 2879 candidates, and accepted 11517 as neighbouring PAGs, out of which 130 (1.1%) were found to be invalid at the final MAG validation check. For the same batch, extended GPS version rejected 5969 candidates at the initial check, while accepting 47523, out of which 2087 (4.4%) were found to be invalid after all.

That means the basic validity tests already filter out close to 95% of all invalid operators. Undoubtedly this can be increased further, but there is a risk the added overhead of significantly more complicated validity tests may not outweigh the benefits of avoiding an extra MEC-PAG-MAG conversion for 1% of the candidates. Similarly, the extended version now captures about 75% of all invalid operators, but this can likely be brought to around the level of the baseline version by adding explicit basic validity checks for each candidate considered by an operator (rather than the single check per operator it is now).

# 7 CONCLUSION

We presented GPS, the first score-based equivalence search algorithm in the presence of latent confounders. It was based on a new MEC characterization for MAGs that brings establishing Markov equivalence between sparse graphs down to linear complexity, with the new core PAG providing the crucial link to efficient PAG reconstruction. Experimental results confirmed our hopes/expectations that equivalence search could traverse the MAG space faster than single-edge MAG modifications, while arriving at better models, comparable to or improving on other state-of-the-art methods, and that additional gains can be expected by incorporating more comprehensive search strategies like tabu-search and multiple restarts. Looking forward, we aim to expand GPS further by considering the full PAG neighbourhood as candidates (similar to GES), and including a more robust equivalence score that can also handle selection bias.

### Acknowledgements

We thank anonymous reviewers for valuable feedback and helpful suggestions.

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
