# OpenReview forum: "Greedy Equivalence Search in the Presence of Latent Confounders"
_auai.org/UAI/2022/Conference — UAI 2022 Poster_

### Official Review · Reviewer_KSBX · 2022-04-10

**Q2(1) Originality/Novelty:** 3
**Q2(2) Significance/Impact:** 3
**Q2(3) Correctness/Technical Quality:** 4
**Q2(6) Clarity Of Writing:** 4
**Q6 Overall Score:** 4
**Q8 Confidence In Your Score:** 5

**Q1 Summary And Contributions:**

The authors propose a score-based greedy search for causal graphs with latent confounders. Along the way, they propose a new characterization of equivalence between MAGs that uses what is called MEC and the proposed greedy algorithm traverses this MEC space.

I read the paper very carefully and enjoyed it for the most part. For that, I would like to thank the authors.

**Q2 Assessment Of The Paper:**

More detailed information regarding each of these aspects is given below:

**Q2(4) Quality Of Experiments (Optional):**

3: Good: The experimental evaluation is adequate, and the results convincingly support the main claims.

**Q2(5) Reproducibility:**

4: Excellent: Key resources (e.g., proofs, code, data) are available and key details (e.g., proof sketches, experimental setup) are comprehensively described for competent researchers to confidently and easily reproduce the main results.

**Q3 Main Strengths:**

MEC characterization is interesting and is new. The proofs seem correct. Score-based discovery with latent variables is a very important research direction.

**Q4 Main Weakness:**

The role of the new characterization in the new algorithm is not very clear (more below). The score function is missing unless the underlying data is Gaussian, which takes away from the contribution. Therefore the proposed algorithm is incomplete for arbitrary distributions.

**Q5 Detailed Comments To The Authors:**

My TL;DR: 1. Good attempt at an important and challenging problem. But falls short of providing a complete solution where an important part of the algorithm is missing unless the data is Gaussian. 2. The proposed new characterization and MEC structure look interesting but the advantage over the MAG architecture is not very clear since the proposed algorithm does not make the best use of this new characterization. The article builds up very nicely and increases the expectation all the way until the actual algorithm is given which doesn't deliver up to that promise in my opinion: If I have to anyway check the validity of each MEC after a graph operation, I could also do that by staying in the MAG domain. Why do we expect MECs to provide a significant gain? I would be happy to hear from the authors in case I am missing something here. I think adding some graphs and traces of the proposed algorithm vs. GES would help demosntrate the advantage of lifting to the MEC domain.

DETAILED COMMENTS:
(resp.) condition is missing from part 2 in the last line of Definition 2.

Proof of Lemma 2 looks correct. Corollary 3 is straightforward and makes sense.

I don't see proof for the correctness of Algorithm 1. please add this. for example, why is this greedy process of adding ordered triples sufficient to add all ordered triples? The proof is needed (I understand it's a simple recursion argument but still).

MEC to MAG construction through core PAG makes a lot of sense. Proof of it (proof of Lemma 4) seems sufficient. I would perhaps emphasize why there are no missed discriminating paths by the proposed construction algorithm.

"Note that the orientation rules also remain sound for arbitrary subsets"
Where will this statement be used later on? Why is it critical to bring up here? Hard to place in the narrative.

"With appropriate indexing that implies an additional d candidates to check"
Could you more explicitly describe this indexing?

"Each entry in the temporary list is then processed and com- pared against d other candidates, each of which can be handled in constant time as it involves only verifying pres- ence in one of the non/collider triple lists, which can again be done in constant time using appropriate indexing, and the presence of a specific edge in G, also in constant time"
I am guessing there is some hashing idea underneath this paragraph but it needs to be made explicit. I would normally expect checking membership in a list to take linear in the size of the list - with no structure.

"Therefore we can move directly between equivalence classes by making single element changes to the MEC, provided that these still correspond to a valid MAG/MEC."
It sounds like one can only make such changes in one direction: From higher order to lower order triples. Is that correct? Why is this not utilized in the later algorithm?

"It means ‘neighbouring’ MECs can be very different, allowing for large(r) steps to be taken, which should help avoid getting stuck in local optima."
This sounds a bit speculative. At the end of the day, you are still taking one step in the MEC step since all those other changes are necessary to arrive at a valid MEC.

I am curious about the following:
if you changed the MakeCollider(x,y,z) rule and oriented x*->z<-*y even for D_{k>=1}, would it still not be sound? I believe currently authors are trying to ensure each collider with order > 0 corresponds to a single edge addition but I am not sure if this was necessary or if it is why. I would appreciate if they can clarify.

"Similarly, MakeNoncollider"
This line may have a problem. Because the suggested change would lead to x*-z-*y which can't happen since it is assumed that there is no selection bias. Please check.

Proof of Theorem 1 seems quite straightforward but also not insightful on whether the proposed search is efficient.

More importantly, the proposed search procedure seems to be relying on trying every possible graph modification and for every such option, go through a lengthy validation step. This sounds like it could be very time-consuming. Isn't there a way to decide in advance which graph modifications are allowed and which aren't? I would have expected the proposed alternative characterization to pave the way for this but after reading the paper I don't think it helps. I do understand though it still provides some experimental improvement as given in Table 3.

Section 5: "For simplicity, in this article we will assume a multivariate Gaussian model."
This is a strong and central assumption - otherwise there is no algorithm without a proper score. Therefore it has to be at the front and center of the manuscript and not appear only in Section 5. Please add this assumption to abstract and introduction. I understand the previous sections are generic but then without an appropriate score, the proposed solution will be incomplete for non-Gaussians.

This issue of having to restrict to Gaussian distributions also makes the contribution over the existing work less clear.

"In particular we want to mention current work on incorporating a general equivalence score that also handles selection bias and cyclic interactions."
The reviewers cannot validate/evaluate a work in progress which makes this statement not very useful. I suggest avoiding this type of statement in the submission.

minor comments:
"we we" on pg 3
"(non-)collider" "non/collider" please pick one not'n.


**Q7 Justification For Your Score:**

Please see TL;DR above.

**Q9 Complying With Reviewing Instructions:**

1: Yes.

---

### Official Review · Reviewer_RMFc · 2022-04-11

**Q2(1) Originality/Novelty:** 3
**Q2(2) Significance/Impact:** 3
**Q2(3) Correctness/Technical Quality:** 2
**Q2(6) Clarity Of Writing:** 3
**Q6 Overall Score:** 5
**Q8 Confidence In Your Score:** 3

**Q1 Summary And Contributions:**

The paper presents a novel greedy PAG search (GPS) algorithm for score-based causal discovery in data with latent confounders. GPS uses a simple new MEC characterization for MAGs for efficient traversal over equivalence classes in the space of all MAGs.
The experiments show the promising performance of GPS w.r.t. the speed and accuracy.


**Q2 Assessment Of The Paper:**

More detailed information regarding each of these aspects is given below:

**Q2(4) Quality Of Experiments (Optional):**

3: Good: The experimental evaluation is adequate, and the results convincingly support the main claims.

**Q2(5) Reproducibility:**

3: Good: Key resources (e.g., proofs, code, data) are available and key details (e.g., proofs, experimental setup) are sufficiently well-described for competent researchers to confidently reproduce the main results.

**Q3 Main Strengths:**

The developed GPS is a novelty and significant method in causal discovery.
The novel MEC characterization for MAGs can be very useful in practice.
The theoretical results and empirical results are clear to show the performance of GPS.

**Q4 Main Weakness:**

*Some definitions are not clear and the details are summarised in Q5.

**Q5 Detailed Comments To The Authors:**

 * In definition 1, "if a *-* b *-* c" is in $\mathcal{G}$, with a and b not adjacent" -> it seems to be "a and c not adjacent" since a and b are trivial adjacent.
* In definition 1, $<q_{p-1}, q_p, b>$ seems to be wrong, and it should be $<q_{p-1}, q_p, a>$.
* The discriminating path is not defined. By the way, the discriminating path requires each collider on this path to be a parent of c.
* In definition 2,  defined recursively as a - b - c is an unshielded collider in $\mathcal{G}$.  a - b - c is a collider in $\mathcal{G}$. It would be better to use the edge with an arrow since $\mathcal{G}$ is a MAG.
* On page 3, after Definition 2, there is an issue with single quotes: 'triple with order'.

**Q7 Justification For Your Score:**

The contributions of the paper seem good and useful, but the current version is not readable even with some errors.

**Q9 Complying With Reviewing Instructions:**

1: Yes.

---

### Official Review · Reviewer_JEB6 · 2022-04-11

**Q2(1) Originality/Novelty:** 2
**Q2(2) Significance/Impact:** 2
**Q2(3) Correctness/Technical Quality:** 3
**Q2(6) Clarity Of Writing:** 3
**Q6 Overall Score:** 4
**Q8 Confidence In Your Score:** 4

**Q1 Summary And Contributions:**

The paper proposes a new characterization of the MEC of MAGs. Then the paper develops a greedy learning scored based algorithm based on it.

**Q2 Assessment Of The Paper:**

More detailed information regarding each of these aspects is given below:

**Q2(4) Quality Of Experiments (Optional):**

3: Good: The experimental evaluation is adequate, and the results convincingly support the main claims.

**Q2(5) Reproducibility:**

2: Fair: Key resources (e.g., proofs, code, data) are unavailable but key details (e.g., proof sketches, experimental setup) are sufficiently well-described for an expert to confidently reproduce the main results.

**Q3 Main Strengths:**

The author proposes a greedy learning score-based algorithm that explores in the space of Markov equivalence class while previous score based algorithms searchs in the space of MAGs. It simplifies Ali's characterization of MEC and results in faster computation time. And their algorithm run faster than other common baseline learning algorithms.

**Q4 Main Weakness:**

Both the new characterization or the method traversing the MEC both rely much on previous work (Ali et al. [2009] and Zhang [2008]), the theory are not difficult and innovative enough. And in terms of measures like BIC/accuracy, the performance are not significantly better than the baseline algorithm.

The paper also didn't analyse the complexity of learning algorithm enough but only the empirical performance.

 Also I'm not sure if the proof of Theorem 1 is valid.

**Q5 Detailed Comments To The Authors:**

(1) For the example in Figure 1, should the collider (C,D,E) have order 2?

(2) For the proof of Theorem 1, I'm not sure if the last sentence is correct. I understand that the operators in algorithm 3 allow changing from a MAG G1 to another MAG G2 with any skeleton and no unshielded collider by iteratively turning unshielded colliders to non-colliders one by one then remove/add edges. However, I'm not sure that, to go from G2 to a MAG G3 with the same skeleton, their operators are sufficient, because they didn't prove that changing one non-collider/collider with order at each step results in a valid MAG.

(3) Without proving meek's conjecture, Theorem 1 is not sufficient to guarantee output the optimal graph.

(3) In simple scenario, like n=5, have the authors checked that the failure of the algorithm is only due to unstable BIC or fail to propose the right neighbor MEC?

(4) Some complexity seems not clearly analysed.Like:
     (a) complexity for verifying valid MEC;
     (b) what's the chance for proposing an invalid MEC or proposing the same MEC multiple times.

**Q7 Justification For Your Score:**

The overall idea is good and new but the theory seems not innotive enough. The complexity are not analysed enough. The empirical performance seems not significantly better compared to other baseline algorithms.

**Q9 Complying With Reviewing Instructions:**

1: Yes.

---

### Official Review · Reviewer_dmny · 2022-04-12

**Q2(1) Originality/Novelty:** 3
**Q2(2) Significance/Impact:** 2
**Q2(3) Correctness/Technical Quality:** 3
**Q2(6) Clarity Of Writing:** 2
**Q6 Overall Score:** 7
**Q8 Confidence In Your Score:** 2

**Q1 Summary And Contributions:**

This paper proposes a method of greedy equivalence search that estimates causal structure in the presence of latent confounders. The authors call this method Greedy PAG Search (GPS). Until now, constraint-based methods have been able to estimate causal structures in the presence of latent confounders, but the method proposed in this study is the first to do so for GES.

**Q2 Assessment Of The Paper:**

More detailed information regarding each of these aspects is given below:

**Q2(4) Quality Of Experiments (Optional):**

2: Fair: The experimental evaluation is weak: important baselines are missing, or the results do not adequately support the main claims.

**Q2(5) Reproducibility:**

3: Good: Key resources (e.g., proofs, code, data) are available and key details (e.g., proofs, experimental setup) are sufficiently well-described for competent researchers to confidently reproduce the main results.

**Q3 Main Strengths:**

This paper is the first contribution to propose a method for estimating causal structures with latent confounders in a greedy equivalence search. This method can speed up the computation time compared to previous constraint-based methods. This method may be extended to models with cycles and selection bias in the future.

**Q4 Main Weakness:**

As stated by the authors, the accuracy of the method proposed in this paper is not superior to that of existing methods. In addition, the authors mention that the GPS algorithm can be used for models with selection bias and cycles, but it is still unclear what impact it can have on this area of research in the future.

**Q5 Detailed Comments To The Authors:**

Please explain why the GPS algorithm is not superior to the existing methods in regard to the accuracy.

Please add an explanation about what impact GPS can have on this area of research in the future in detail.

**Q7 Justification For Your Score:**

Despite some weaknesses in this paper, there has been no previous GES-based method for estimating causal models with latent confounders, and this paper should serve as a catalyst for further research.

**Q9 Complying With Reviewing Instructions:**

1: Yes.

---

### Official Review · Reviewer_kX5F · 2022-04-12

**Q2(1) Originality/Novelty:** 3
**Q2(2) Significance/Impact:** 2
**Q2(3) Correctness/Technical Quality:** 3
**Q2(6) Clarity Of Writing:** 3
**Q6 Overall Score:** 6
**Q8 Confidence In Your Score:** 4

**Q1 Summary And Contributions:**

This paper introduces a MEC characterization for MAGs, and then proposes a score-based method for causal discovery in the presence of latent confounders, named Greedy PAG Search (GPS) algorithm. Theoretical analysis guarantees the correctness of the algorithm, and experiment results verify the effectiveness of the algorithm.

**Q10 Ethical Concerns (Optional):**

No.

**Q2 Assessment Of The Paper:**

More detailed information regarding each of these aspects is given below:

**Q2(4) Quality Of Experiments (Optional):**

2: Fair: The experimental evaluation is weak: important baselines are missing, or the results do not adequately support the main claims.

**Q2(5) Reproducibility:**

2: Fair: Key resources (e.g., proofs, code, data) are unavailable but key details (e.g., proof sketches, experimental setup) are sufficiently well-described for an expert to confidently reproduce the main results.

**Q3 Main Strengths:**

1) This paper exploits the information of PAG to search for the best Markov equivalence class (MEC) of Maximal ancestral graphs, which makes the proposed algorithm more efficient. This is also confirmed by the experiment results.
2) The author proposes a new characterization of MEC, and provides some theories to ensure the completeness of the output results.


**Q4 Main Weakness:**

1) In Algorithm 3, the authors mentioned a function `PAG_to_MEG()`, but I could not find how to execute this procedure. It would be better to provide details about `PAG_to_MEG()`.
2) Several score-based methods are proposed to recover causal graphs in the presence of latent confounders, but they are not discussed in the paper.

**Q5 Detailed Comments To The Authors:**

Strengths:
1) This paper exploits the information of PAG to search for the best Markov equivalence class (MEC) of Maximal ancestral graphs, which makes the proposed algorithm more efficient. This is also confirmed by the experiment results.
2) The author proposes a new characterization of MEC, and provides some theories to ensure the completeness of the output results.


Weakness:
1) In Algorithm 3, the authors mentioned a function `PAG_to_MEG()`, but I could not find how to execute this procedure. It would be better to provide details about `PAG_to_MEG()`.
2) The author mentioned the discriminating path rules `R4a/b in [Zhang, 2008]` in Section 3.4, but I only find the `R4` in [Zhang, 2008]. I think it is better to clarify what `R4a` and `R4b` are.
3) Several score-based methods [1,2] are proposed to recover causal graphs in the presence of latent confounders, but they are not discussed in the paper. Specifically, the M3HC algorithm [1] aims to recover MAG. It would be better to compare the proposed method against the M3HC algorithm in the experiment.

[1]Tsirlis K, Lagani V, Triantafillou S, et al. On scoring maximal ancestral graphs with the max-min hill climbing algorithm. International Journal of Approximate Reasoning, 2018, 102: 74-85.
[2]Bhattacharya R, Nagarajan T, Malinsky D, et al. Differentiable causal discovery under unmeasured confounding. International Conference on Artificial Intelligence and Statistics. PMLR, 2021: 2314-2322.

Some minor errors:
1.	The way of citing papers somewhere is not appropriate, e.g., `models Richardson and Spirtes [2002]` should be `models [Richardson and Spirtes 2002]` (in Section 5.1). It would be better to proofread before submitting the paper.

2.	Typos:
-	In Definition 1, `with a and b` -> `with a and c`;
-	In the first sentence of Section 3.3, `we we …` -> `we …`;
-	There is a period missing after the statement of Theorem 4.


**Q7 Justification For Your Score:**

Inferring causal discovery in the presence of latent confounders is a very difficult but promising problem. Although there are some scored-based methods proposed to solve this problem, they only use the information of MAG, which have high time complexity. In this paper, the author explores the Markov equivalence class characterization for MAGs, which makes the proposed algorithm not only ensures the correctness of the algorithm but also makes the algorithm run faster.

**Q9 Complying With Reviewing Instructions:**

1: Yes.

---

### Decision · Program_Chairs · 2022-05-15

**Decision:**

Accept (Poster)

**Comment:**

Meta Review: This a nice paper that makes a real contribution to the literature on Markov equivalence in ancestral graphs in two ways.  First, they introduce the 'core PAG', which is a less oriented version of the full PAG.  This enables their second contribution, which is a faster way to test Markov equivalence of Maximal Ancestral Graphs.  They develop this into a search algorithm they call GPS ('Greedy PAG Search'), analogous to GES for DAGs.  The main weakness is that the explanation for why one should want to use PAGs for search is not made very precise by the authors.

### Minor Comments
 - below table 1: inverted comma on `triple with order';
 - Definition and Lemma 4 - missing period at the end;
 - page 4: "a'core'" $\to$ "a 'core'";
 - page 5: "perform very similar" $\to$ "perform very similarly";
 - There is inconsistency in whether (e.g.) 'definition' is capitalized throughout the manuscript.  Please make this consistent.
 - Capitalization in the references is non-existent.

Another problem is that the simulations are not replicable.  You must specify in the camera-ready version exactly how the parameters were generated for each of your models, as this is not stated anywhere in the submitted manuscript.